# Incorporating Foreshocks in an Epidemic-like Description of Seismic Occurrence in Italy

Giuseppe Petrillo [1] and Eugenio Lippiello [2,*]

1   The Institute of Statistical Mathematics, Research Organization of Information and Systems, Tokyo 190-0014, Japan
2   Department of Mathematics and Physics, University of Campania "L. Vanvitelli", 81100 Caserta, Italy
*   Correspondence: eugenio.lippiello@unicampania.it

**Abstract:** The Epidemic Type Aftershock Sequence (ETAS) model is a widely used tool for cluster analysis and forecasting, owing to its ability to accurately predict aftershock occurrences. However, its capacity to explain the increase in seismic activity prior to large earthquakes—known as foreshocks—has been called into question due to inconsistencies between simulated and experimental catalogs. To address this issue, we introduce a generalization of the ETAS model, called the Epidemic Type Aftershock Foreshock Sequence (ETAFS) model. This model has been shown to accurately describe seismicity in Southern California. In this study, we demonstrate that the ETAFS model is also effective in the Italian catalog, providing good agreement with the instrumental Italian catalogue (ISIDE) in terms of not only the number of aftershocks, but also the number of foreshocks—where the ETAS model fails. These findings suggest that foreshocks cannot be solely explained by cascades of triggered events, but can be reasonably considered as precursory phenomena reflecting the nucleation process of the main event.

**Keywords:** statistical seismology; numerical modeling; probabilistic forecasting; time-series analysis

## 1. Introduction

The Epidemic Type Aftershock Sequence (ETAS) model is widely regarded as the gold standard for seismic predictions and validating hypotheses related to seismic clustering [1–5]. In this model, the increase in seismic activity immediately after the mainshock is attributed to a "bottom-up" triggering process [6]. Essentially, any earthquake can generate a certain number of aftershocks, which are typically of smaller magnitudes, but there is also a non-negligible chance that it could trigger a larger magnitude earthquake. In the latter case, the seismic rate increases before the occurrence of the mainshock, and earthquakes responsible for this increase are referred to as foreshocks. While foreshocks originating from a cascade of triggered events, as in the ETAS model, follow the same patterns as aftershocks and are therefore not particularly informative for predicting large earthquakes, there is a possibility that foreshocks are due to the cumulative effects of tectonic loading processes on the fault that will host the mainshock [7,8]. This effect is commonly referred to as "top-down" loading, and through this mechanism, foreshocks can act as passive tracers of the preparatory process for the impending mainshock. Therefore, in principle, foreshocks can be used to improve predictions about mainshock occurrence.

The debate about the origin of foreshocks and their prognostic value is still ongoing. However, much effort has been made on both sides to extract useful information from instrumental catalogs. Supporting the bottom-up triggering process, several research lines have indicated that the Epidemic Type Aftershock Sequence (ETAS) model could explain the most relevant statistical features of foreshocks [9–12]. Marzocchi and Zhuang [13] have also shown that foreshock activity observed in the Italian catalog is consistent with what is expected by the ETAS model, and they attribute the variability of the statistical features

of foreshocks to the limited sample size. On the other hand, supporting the top-down loading process, Brodsky [14] documented a lack of foreshocks in the ETAS synthetic catalog. Other studies [15–20] have confirmed that the number of foreshocks predicted by the ETAS model is less than the number observed in the instrumental catalog. In order to solve the insufficiencies in foreshock predictions in the ETAS catalog, Petrillo and Lippiello [21] introduced the ETAFS model, integrating both the aftershock and foreshock phenomena. By incorporating this innovative framework, the ETAFS model promises a new description of seismicity patterns and provides us with new insights into earthquake prediction and hazard assessment. This model is in good agreement with the number of foreshocks and aftershocks observed in the Southern California instrumental catalog. This perspective is supported by recent mechanical models of seismic faults, such as those proposed in ref. [22–27]. These models are a generalization of the original spring-block model [28,29] and present seismic patterns with the occurrence of the largest earthquakes frequently preceded by smaller foreshocks. This paper extends the work of [21] to the Italian seismic catalog. We improve the ETAS model by first accounting for the aftershock incompleteness present in the experimental catalog, which is caused by the overlapping of the coda waves [30,31]. We then add the ingredient of foreshocks, which represents a conceptual change for the model. By preserving the point process nature of the model, we replace the occurrence probability of a single earthquake with the occurrence probability of a cluster of earthquakes, which is composed of an earthquake anticipated by its own foreshocks. As for aftershocks, the number of events in each cluster depends on the magnitude of the final earthquake, and their space-time occurrence depends on the space-time of the final earthquake in the cluster. After defining the ETAFS model, we conduct rigorous statistical tests for the aftershock and foreshock numbers for both the ETAS and ETAFS models.

The paper is structured as follows. In Section 2, we introduce the definitions of main-shock, aftershock, and foreshock, and discuss declustering techniques. In Section 3, we present the various models under consideration. Section 4 explains the statistical validation method and the null hypothesis. We present our results in Section 5, followed by the conclusions in the final section.

## 2. General Definitions and Declustering

*Aftershock, Foreshocks and Declustering Techniques*

There are several ways to decluster a numerical earthquake catalog, resulting in different definitions of aftershocks, foreshocks, and mainshocks. In this paper, we consider the Baiesi–Paczusky–Zaliapin–Ben-Zion (BPZB) declustering method proposed in [32–35], which relies on a metric to quantify the correlation between events. The nearest neighbor of an event is defined by the metric $\eta_{ij} = t_{ij} r_{ij}^d 10^{-b(m_i - m_c)}$, where $t_{ij}$ is the time difference between events $i$ and $j$, $r_{ij}$ is their epicentral distance, $d$ is the fractal dimensionality of epicenters, $b$ is the exponent of the Gutenberg–Richter (GR) law, $m_i$ is the magnitude of the first event, and $m_c$ is the completeness magnitude. Two earthquakes form a cluster if their distance $\eta_{ij}$ is below a certain threshold $\eta_c$. To determine $\eta_c$, we define the magnitude-normalized time and space components as follows

$$\tau_{ij} = t_{ij} \times 10^{-bm_i/2} \qquad s_{ij} = r_{ij} \times 10^{-bm_i/2} \tag{1}$$

and for any pair of events we plot $log(s_{ij})$ vs. $log(\tau_{ij})$ in Figure 1. We use $d = 1.6$ for the fractal dimensionality, and $b = 0.95$ obtained from the magnitude distribution (Figure 2). Similar results are obtained for similar choices of $b$ and $d$. In Figure 1, two distinct populations can be observed: the foreshock/aftershock clustering, consisting of events with $s_{ij}\tau_{ij} < \eta_c$, and the stationary Poisson seismicity, consisting of events $i$ with $s_{ij}\tau_{ij} > \eta_c$ for any $j$. The red line in Figure 1 represents the clear boundary between these two populations, which is achieved with $\eta_c = 10^6$. For each event $i$ in the Poisson population, we identify a seismic sequence that includes event $i$ and all $k$ clustered events with $\eta_{ki} < \eta_c$. The largest event in the sequence is defined as the mainshock, while events occurring

before the mainshock are classified as foreshocks and those occurring after are classified as aftershocks.

To avoid obscuring the observation of foreshocks, we apply an additional filter proposed by [19] to the declustering procedure. Specifically, if an event *i* is identified as a mainshock after the BPZB procedure, but it occurs close enough in time and space to a previous large earthquake ($m \geq 5$), then it is considered an aftershock of the earlier event rather than a background event. More precisely, if the probability that the *i*-event is triggered by the $m \geq 5$ earthquake is higher than the background probability $\mu$, the entire cluster is excluded from the study.

After identifying the complete cluster of seismic events, we group all mainshocks into magnitude classes $m \in [m_M, m_M + 1)$, and for each aftershock and foreshock, we define a temporal distance $\Delta t_M$ and an epicentral distance $\Delta r_M$ from the corresponding mainshock. We then define $n_A(T, R, m_L, m_M)$ and $n_F(T, R, m_L, m_M)$ as the total number of aftershocks (and foreshocks) with $\Delta t_M < T$, $\Delta r_M < R$, and magnitude $m \in [m_L, m_L + 0.5)$ linked to a mainshock with magnitude $m \in [m_M, m_M + 1)$.

In this study, we use $T = 10$ days, $m_M = 4, 5$, $m_L = 2, 2.5, 3, 3.5$, and $R = L(m_M) = 0.01 \times 10^{0.5m_M}$.

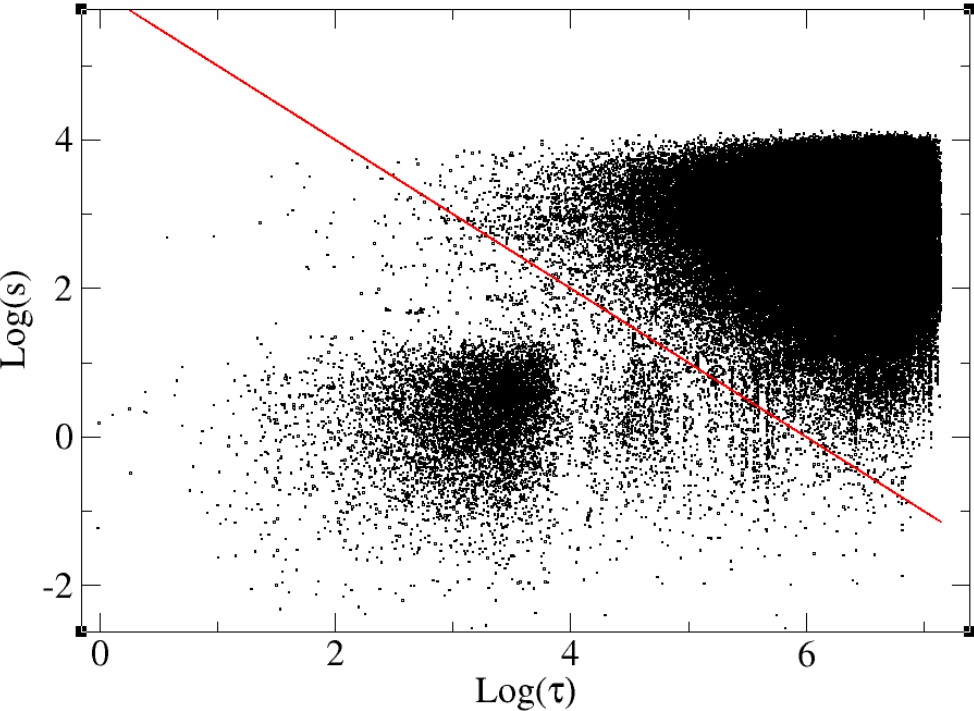

**Figure 1.** Bimodal distribution of time and space components of the nearest-neighbor for the observed seismicity in Italy. Solid red line corresponds to $log_{10}(s) + log_{10}(\tau) = 6$. The fractal dimensionality is fixed at $d = 1.6$.

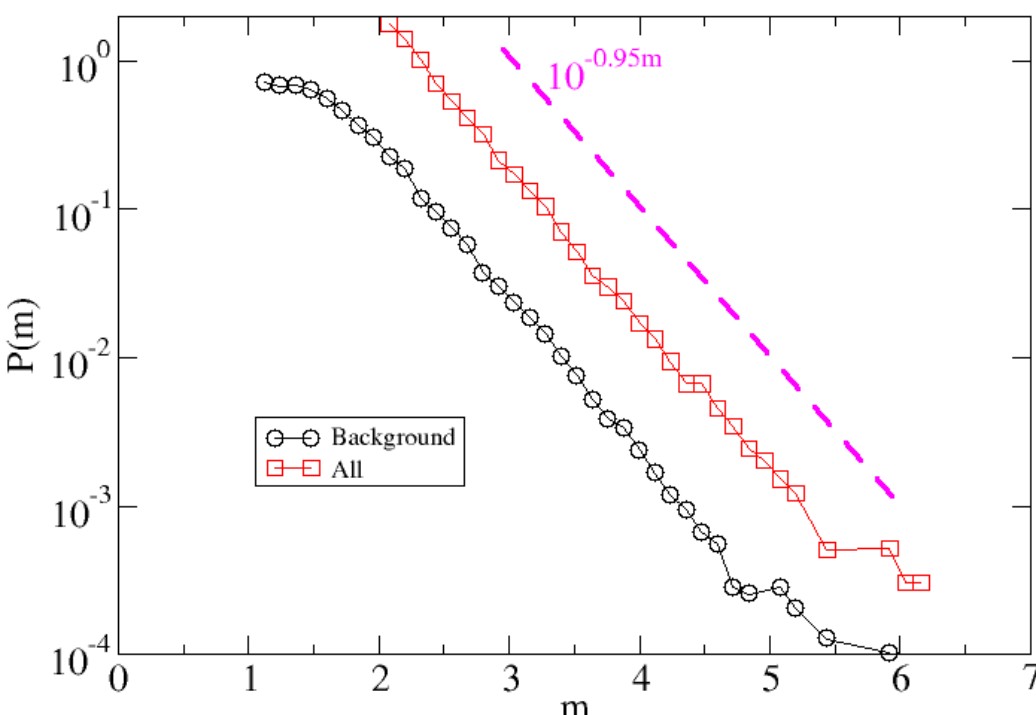

**Figure 2.** Instrumental magnitude distribution $P(m)$ for the Italian seismicity. Black circles represents only the background activity whereas red squares the whole catalog.

## 3. The Models

### 3.1. The Fixed-α ETAS Model

In the Epidemic Type Aftershock Sequence (ETAS) the occurrence rate $\lambda$ of an event with magnitude $m > m_0$, at the position $(x, y)$, at a time $t$ is given by

$$\Lambda(m, x, y, t) = \mu(x, y) + \sum_{j:t_j < t} K10^{\alpha(m_j - m_0)} \frac{(p-1)c^{p-1}}{(t - t_j + c)^p} \frac{q-1}{\pi} (\delta(m_j))^{q-1} [(x - x_j)^2 + (y - y_j)^2 + \delta(m_j)]^{-q} \quad (2)$$

with $\delta(m_j) = d10^{\gamma(m_j - m_0)}$. In the fixed-$\alpha$ ETAS model, $\alpha = b$, a choice which reduces the incompleteness of the seismic catalog [19,36] and take into account spatial anisotropy [37,38] of the seismic events.

### 3.2. The Incomplete ETAS Model: The ETASI Model

A crucial prerequisite for an accurate estimation of the ETAS parameters is the completeness of the seismic catalog. Unfortunately, due to the strong temporal correlation between earthquakes, there is a significant lack of events, particularly in the immediate aftermath of high-magnitude mainshocks [39–43]. This is caused by the overlapping of coda waves, which is even more pronounced when focusing on larger earthquakes where aftershocks are more challenging to detect and report in experimental catalogs [30,44–48]. To address this issue, we introduce the concept of short-term aftershock incompleteness into the ETAS model, resulting in the ETAS Incomplete (ETASI) model [30]. The core idea is that the probability of detecting an event at time $t - t_i$ after an $m_i$ earthquake depends on the difference between its magnitude $m$ and the detection threshold $m_i^{th}(t - t_i)$. Specifically, we adopt the functional form proposed by [39,40].

$$m_i^{th}(t - t_i) = m_i - w \log(t - t_i) - \delta_0 \quad (3)$$

where $w$ and $\delta_0$ are two parameters obtained to reach the best agreement with the instrumental catalog. The functional form for $m_i^{th}$ (Equation (3)) is the most compatible with experimental data, and its logarithmic decay can be explained by the behavior of the seismic

waveform envelope $\mu(t)$ after a mainshock [44,49]. This envelope is always greater than a minimum value $\mu_c(t)$, which decays logarithmically. Lippiello et al. [44] have linked the existence of $\mu_c(t)$ to the overlap between aftershock coda waves, and demonstrated that its decay incorporates parameters related to the Omori–Utsu law governing the decay of aftershocks [50]. Therefore, the expected number of aftershocks in the immediate aftermath of a mainshock can be estimated [49]. We implement the expression Equation (3) in the ETAS model by means of the function $\Phi(m|m_j^{th}(t-t_j),\sigma)$ which represents the cumulative distribution of the normal distribution. In particular, $\Phi$ is a decreasing function of $m$ which is roughly equal 1 when $m > m_j^{th}(t-t_j)+\sigma$, whereas $\Phi \simeq 0$ when $m < \lambda - \sigma$.

The function $\Phi$ can be implemented in the ETAS model Equation (2) as

$$\Lambda(m,x,y,t) = \Lambda(x,y,t) \times \prod_{j:t_j<t} \Phi(m|m_j^{th}(t-t_j),\sigma) \tag{4}$$

The implementation of the ETASI model is straightforward, as it involves generating a synthetic ETAS catalog and then applying the $\Phi$ function to remove a portion of the events. The parameters of the ETASI model include all those of the ETAS, as well as two additional parameters that account for the incompleteness function. However, we do not restrict our analysis to fitting only the last two parameters ($w$ and $\delta_0$); rather, we perform a global optimization of all 10 parameters in the model.

### 3.3. The Top-Down Loading ETAS Model—The ETAFS Model

By construction, the previously defined models assume bottom-up triggering as the hypothesis for foreshocks. In other words, a foreshock is considered a normal earthquake that triggers an offspring with a magnitude greater than itself. However, it is important to also consider the possibility of top-down loading as an explanation for foreshocks. The Epidemic Type Aftershocks and Foreshocks Sequence ETAFS model is a generalization of the ETAS model that incorporates the mechanism of loading by aseismic slip to account for top-down triggering. In the ETAFS model, aftershocks are triggered with the same probability rate as the ETAS model, but each earthquake can also be preceded by foreshock activity. Mathematically, this can be expressed as:

$$\Lambda_{ETAFS}(m,x,y,t) = \Lambda_{ETAS} + \sum_{k} 10^{-bm} Q_f(d_{ik},t_k-t,m_k) \tag{5}$$

where $d_{ik}$ is the distance between the event $i$ and $k$ and the sum extends over all events with magnitudes $m_k$, occurred at time $t_k$, in the position $(x_k,y_k)$ triggered according to the ETAS probability (Equation (2)), whereas $Q_f(d_{ik},t_k-t,m_k)$ is the rate of foreshocks potentially occurring at times $t < t_k$. There is no specific constraint on the functional form of $Q_f$. In order to reduce the number of model parameters we assume [21] that the spatio-temporal organization of the event in the cluster is similar to the aftershock one, setting

$$Q_f(d_{ik},t_k-t,m_k) = K_f 10^{\alpha_f(m_k-m_0)} \frac{(p_f-1)c_f^{p_f-1}}{(t_k-t+c_f)^p} \frac{q_f-1}{\pi} (\delta(m_k))^{q_f-1} \left(d_{ik}^2+\delta(m_k)\right)^{-q_f} \tag{6}$$

where $\delta(m_k) = d_f 10^{\gamma_f(m_k-m_0)}$, $\gamma_f = \gamma$, $d_f = d$, and $q_f = q$. As in the ETAS model, here we extract the number of events belonging to $k$-th cluster from a Poissonian distribution with average $K_f 10^{\alpha_f(m_k-m_0)}$. Moreover we implement the inverse Omori Law with the same $p$ as in the aftershock occurrence and $c_f = c$. In principle, different functional forms could be adopted to achieve similar results. Finally, we apply the same aftershock removal procedure as done for the ETAS model in order to take into account the hiding of events caused by the overlapping of coda waves. In mathematical terms

$$\Lambda_{ETAFS}^{inc}(m,x,y,t) = \Lambda_{ETAFS}(m,x,y,t) \times \prod_{j:t_j<t} \Phi(m|m_j^{th}(t-t_j),\sigma) \tag{7}$$

For the simulations of the ETAFS model, we use exactly the same parameters as the ETASI model to generate a complete ETAS catalog. Starting from each simulated ETAS earthquake, we then use the kernel $Q_f$ to generate foreshocks. The magnitude of each foreshock is extracted from the Gutenberg–Richter law, but with the constraint that the magnitude of each event belonging to the cluster must be smaller than the final magnitude $m_k$. We finally apply the filtering procedure by means of the function $\Phi$, according to Equation (7), to take into account incompleteness.

The apparent problem with the ETAFS model is that the spatio-temporal organization of the cluster's events depends directly on the characteristics of the incoming event, the mainshock. In practice, it seems that this model violates the temporal causality principle since the occurrence probability at a certain time depends on the future. However, from the point of view of the point process-like model, the ETAFS remains well defined if one considers each single ETAS event as a cluster of events. In particular, in a formulation practically similar to the ETAS model, the ETAFS model can be viewed as a point process, where each single point has an internal structure represented by an earthquake and its anticipating foreshocks. With this approach, Equation (2) gives the occurrence probability of the last event of the cluster, and all events that belong to the same cluster are deterministically correlated with each other. This reflects the idea that events belonging to the same cluster are the manifestation of the same underlying process and contain information on the incoming event.

We generate synthetic catalogs with the same algorithm used in [21] and the parameter values used in the three models are given in Table 1.

**Table 1.** Parameters for the three models presented in this study. We consider the lower magnitude threshold $m_0 = 2$.

| Model | $K_0$ | $\alpha$ | $b$ | $p$ | $c$ | $d$ | $\gamma$ | $q$ | $\omega$ | $\delta_0$ | $K_f$ | $\alpha_f$ |
|---|---|---|---|---|---|---|---|---|---|---|---|---|
| ETAS | 0.07 | 0.95 | 0.95 | 1.2 | 0.024 | 0.006 | 1.958 | 1.3 | - | - | - | - |
| ETASI | 0.1 | 0.93 | 0.95 | 1.2 | 0.01 | 0.006 | 1.958 | 1.3 | 0.3 | 1 | - | - |
| ETAFS | 0.1 | 0.93 | 0.95 | 1.2 | 0.01 | 0.006 | 1.958 | 1.3 | 0.3 | 1 | 0.05 | 0.5 |

## 4. Data Catalog, Methods and Null Models

The purpose of the statistical test that will be carried out in this paper is to evaluate whether the three models defined in the previous section are able to describe the experimental data.

In the null-hypothesis test we compare the number of aftershocks and foreshocks for each mainshock. In practice we consider the ISIDE Italian Catalog (from 2005/04/16 to 2021/04/30). We use the BPZB selection procedure to identify aftershocks and foreshocks and we evaluate the quantities $n_A(T, R, m_L, m_M)$ and $n_F(T, R, m_L, m_M)$ for a wide range of parameter $m_L = [2, 2.5, 3, 3.5]$, $m_M = [4, 5, 6]$, and for $T = 1, 3, 10$ days, always considering $R = L(m_M)$ km. We compute these quantities for each model we defined before, namely ETAS, ETASI and ETAFS and we consider each of them as null model for the hypothesis test. We choose model parameters such as the models lead to synthetic catalogs of equal duration of the ISIDE catalog and roughly the same number of events with $M > 2.5$. For each model and for a defined set of model parameters, we generate 1000 catalogues that we concatenate and we identify seismic sequences by means of a BPZB procedure. After the identification of the sequences, we divide the whole catalog in $N_s$ subset where each one contains the same number of the earthquakes of instrumental Italian catalogue. Now we compute the quantity $n_{A,j}^X$ and $n_{F,j}^X$, which represents the number of aftershocks and foreshocks, respectively, in the $j$-th subset produced by the $X$ model. The same quantity is computed for the instrumental Italian catalogue. The superscript, $X$, indicates a specific numerical model with the three possible entries $X = [ETAS, ETASI, ETAFS]$, according to the corresponding model. Conversely, we no longer use the superscript $X$ for the number of aftershocks and foreshocks in the instrumental Italian catalogue.

We evaluate the difference $\psi_{j,A} = \frac{n^X_{A,j}-n_A}{n^X_{A,j}+n_A}$ and $\psi_{j,F} = \frac{n^X_{F,j}-n_F}{n^X_{F,j}+n_F}$. This allows us to quantify the difference in the number of aftershocks and foreshocks, respectively, between the *j*-th realization of the synthetic catalogue of the *X* model and the instrumental ISIDE catalogue. In particular we evaluate these quantities for a different sets of parameters $(T, R, m_L, m_M)$. and compute two histograms $H(\psi_{j,A})$ and $H(\psi_{j,F})$ defined as the fraction of the $N_s$ subsets with a value $\psi_A = \psi \pm d\psi$, or $\psi_F = \psi \pm d\psi$, where $d\psi = 0.005$. If the histogram $H(\psi)$ is very peaked around $\psi = 0$, the prediction of the model is in good agreement with the experimental catalogue, i.e., the number of aftershocks (foreshocks) in the model is similar the instrumental one. The two limits $\psi = -1$ and $\psi = 1$ represents the worst cases. In particular, with $\psi = -1$, the synthetic catalogue predicts a number of events equal to 0. The opposite behaviour is observed with $\psi = 1$, which represents the limit for infinite events predicted in the simulated catalogue. For the foreshock statistical evaluation, it is useful to define the quantities $I^X_+ = \sum_{\psi>0} H(\psi_{j,F})d\psi$, representing the area under the histogram for positive $\psi$ values.

The overall procedure is shown in detail in Figure 3.

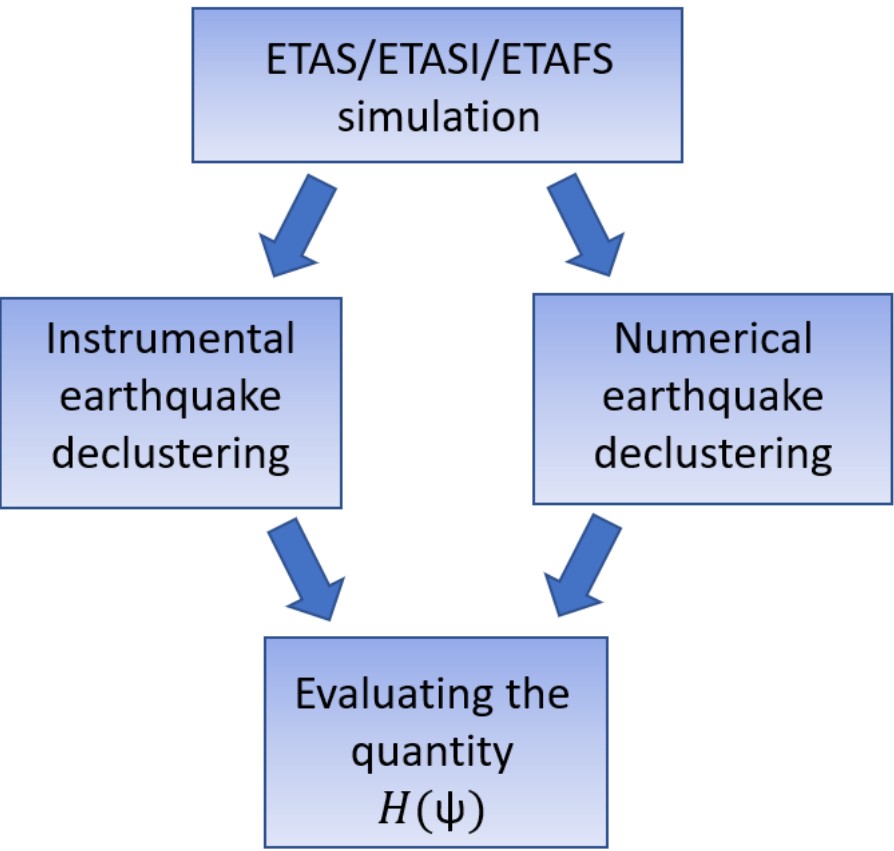

**Figure 3.** Flowchart of the method employed in the study.

## 5. Results

### 5.1. Aftershock Comparison

In this subsection we present results of the comparison for the aftershock number observed in the synthetic catalogs of the ETAS and ETAFS models with those selected in the instrumental catalog. The number of aftershocks in the ETAFS and the ETASI model roughly coincides, therefore we limit ourselves to present results for the ETAFS model.

In Figure 4 the histogram $H(\psi)$ is plotted for the ETAS model and the ETAFS model, considering different values of $m_M$, $m_L$ and different time windows $T$. The best agreement with the instrumental catalog is obtained when $H(\psi)$ is a very peaked distribution around $\psi \simeq 0$. This corresponds to the situation when the majority of synthetic catalogs present

a number of aftershocks similar to the one of the instrumental catalog. This condition is clearly satisfied for $m_M \leq 5$ for both the ETAS and the ETAFS model. Only for $m_M = 5$ and $m_L = 2$, we observe that the maximum of $H(\psi)$ in the ETAS model is located at $\psi \simeq 0.25$ indicating an excess of aftershocks in the ETAS catalog compared to the instrumental one. This shift on the right of peak of $H(\psi)$ can be attributed to the short term aftershock incompleteness, as confirmed by the fact that the peak of $H(\psi)$ in the ETAFS model is close to $\psi = 0$. This shift on the right is still present for $m_m = 5$ and $m_L = 2.5$ but it is less relevant and it disappears by increasing $m_L$, as expected since short-term aftershock incompleteness is less relevant the larger the aftershock magnitude is. The situation for $m_M = 6$ is less clear since the distribution is much broader, both for the ETAS and the ETAFS models, probably because there are only 3 $m > 6$ mainshocks in the instrumental catalog. Nevertheless, we notice that for $m_M = 6$ and $m_L \leq 3$ the distribution $H(\psi)$ for the ETAS model is significantly asymmetric with a clear excess of aftershocks in the ETAS catalog compared to the instrumental catalog. This can be again attributed to short term aftershock incompleteness, which is more relevant for larger mainshocks, and it is confirmed by the observation that $H(\psi)$ for the ETAFS model is more symmetric.

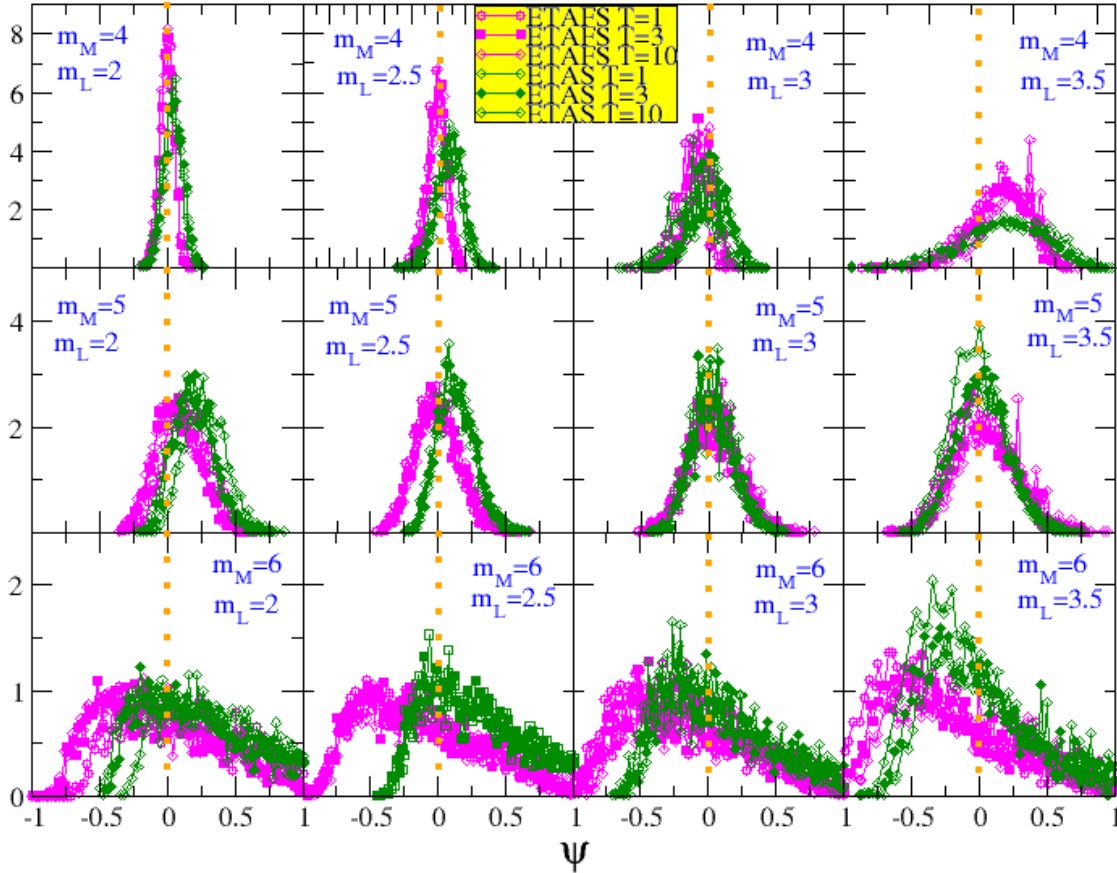

**Figure 4.** The fraction of numerical subsets $H(\psi)$ of the ETAFS (ETASI) and ETAS simulated catalogue, with a number of aftershocks $\frac{n_{A,j}^X(T,R,m_L,m_M) - n_A(T,R,m_L,m_M)}{n_{A,j}^X(T,R,m_L,m_M) + n_A(T,R,m_L,m_M)} = \psi \pm 0.005$. Results are for different values of the time window $T$. Moving horizontally, from left to right, $m_L = 2, 2.5, 3, 3.5$ whereas vertically, from top to bottom, $m_M = 4, 5, 6$. The orange dashed line indicate the optimal description of the seismicity $\psi = 0$.

### 5.2. Foreshock Comparison

In Figure 5 we plot the histograms $H(\psi)$ of $\psi_{j,F}$ for the ETAS and ETAFS catalogues. We will not present results for the ETASI model which are comparable to those of the ETAS model, since the two models only differ in the implementation of short term incompleteness,

which is not relevant for foreshocks. As for the aftershocks, we consider different windows of time $T$, foreshock minimum magnitude $m_L$ and mainshock threshold $m_M = 4, 5$. We will present results for mainshocks with $m > 6$ separately. This figure shows that the ETAS catalogue presents significant less foreshocks than the instrumental Italian one. Indeed $H(\psi)$ is in all cases peaked around $\psi \simeq -0.5$, which indicates that the number of foreshocks in the ETAS catalogue is, on average, about $1/3$ the number found in the Italian catalogue. Moreover, we note that the probability of producing a numerical catalog that has the same number of foreshocks as the observed catalog is practically zero. Indeed, no ETAS synthetic catalog presents a number of foreshocks equal or larger than the one observed in the instrumental catalog when $m_M = 4$ and, at the same time, when $m_M = 5$ less than 1% of synthetic catalogs satisfies this condition. This conclusion can be drawn for all the foreshocks magnitude thresholds $m_L$, and for each time window $T$ considered. This interpretation is supported from the measurement of the quantity $I_+^{ETAS}$ reported in Table 2. Indeed, for all values of $m_M$, $m_L$ and $T$ the value of $I_+^{ETAS}$ is nearly zero. We conclude that it is very unlikely or even impossible that a synthetic catalog for the ETAS model presents a number of foreshocks equal or larger than the one observed in the Italian seismic catalogue. The situation slightly changes when one considers the ETASI model, with $I_+^{ETASI}$ (Table 2) still indicating that the hypothesis that the ETASI model predicts a number of foreshocks equal to or larger than those actually observed can be rejected with a high confidence.

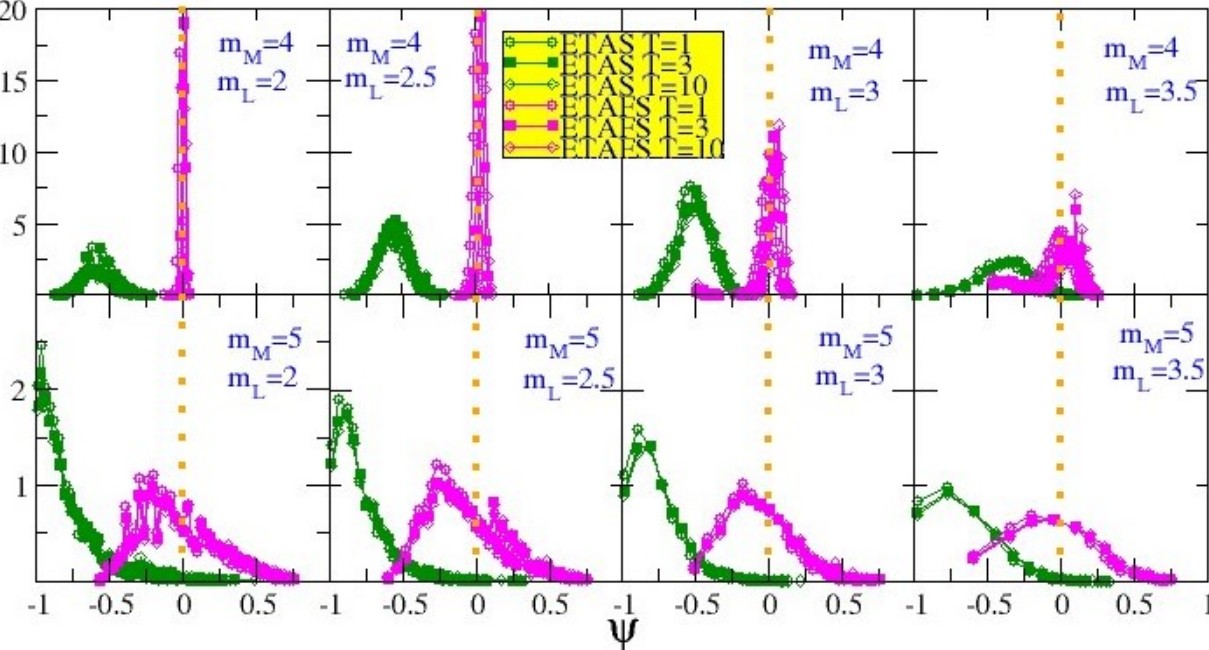

**Figure 5.** The fraction of numerical subsets $H(\psi)$ of the ETAS and ETAFS simulated catalogue, with a number of foreshocks $\frac{n_{F,j}^X(T,R,m_L,m_M) - n_F(T,R,m_L,m_M)}{n_{F,j}^X(T,R,m_L,m_M) + n_F(T,R,m_L,m_M)} = \psi \pm 0.005$. Results are for different values of time window $T$. Moving horizontally from left to right, $m_L = 2, 2.5, 3$, and 3.5, whereas from top to bottom, $m_M = 4$ and 5. The orange dashed line indicates the optimum description of the seismicity $\psi = 0$.

On the contrary, the numerical catalog simulated by means of the ETAFS model captures the seismicity of the foreshocks very well; in fact, all the $H(\psi)$ histograms exhibit a peak that is very close to $\psi = 0$. We observe this behavior for all minimum magnitude $m_L$ and temporal domains $T$. It is worth noting that a dramatic change in $H(\psi)$ is obtained by adding just a small percentage of earthquakes (about 10%) to the ETASI catalog. This small percentage of earthquakes are foreshocks triggered according to the second term in Equation (5), indicating that their addition is strictly necessary to obtain a good agreement with the observed seismicity in the Italian catalogue. This is confirmed quantitatively in

Table 2, where all $I_+^{ETAFS}$ values are significantly larger than 0 and smaller than 1, presenting fluctuations around the optimal value ($I_+^{ETAFS} = 0.5$).

**Table 2.** The positive area below the curve $H(x)$ for ETAS and ETASI model for the foreshocks.

| $m_M$ | $m_L$ | $T(d)$ | $I_+^{ETAS}$ | $I_+^{ETASI}$ | $I_+^{ETAFS}$ |
|---|---|---|---|---|---|
| 4 | 2 | 1 | 0 | 0 | 0.6 |
| 4 | 2 | 3 | 0 | 0.01 | 0.73 |
| 4 | 2 | 10 | 0 | 0.02 | 0.76 |
| 4 | 2.5 | 1 | 0 | 0 | 0.50 |
| 4 | 2.5 | 3 | 0 | 0 | 0.8 |
| 4 | 2.5 | 10 | 0 | 0.01 | 0.85 |
| 4 | 3 | 1 | 0 | 0.01 | 0.43 |
| 4 | 3 | 3 | 0 | 0.03 | 0.80 |
| 4 | 3 | 10 | 0 | 0.05 | 0.92 |
| 4 | 3.5 | 1 | 0 | 0.10 | 0.30 |
| 4 | 3.5 | 3 | 0.02 | 0.10 | 0.51 |
| 4 | 3.5 | 10 | 0.03 | 0.15 | 0.58 |
| 5 | 2 | 1 | 0 | 0.03 | 0.32 |
| 5 | 2 | 3 | 0.01 | 0.05 | 0.39 |
| 5 | 2 | 10 | 0.02 | 0.07 | 0.42 |
| 5 | 2.5 | 1 | 0 | 0.01 | 0.29 |
| 5 | 2.5 | 3 | 0 | 0.03 | 0.37 |
| 5 | 2.5 | 10 | 0 | 0.05 | 0.40 |
| 5 | 3 | 1 | 0 | 0.01 | 0.28 |
| 5 | 3 | 3 | 0 | 0.03 | 0.34 |
| 5 | 3 | 10 | 0 | 0.05 | 0.37 |
| 5 | 3.5 | 1 | 0.01 | 0.16 | 0.22 |
| 5 | 3.5 | 3 | 0.03 | 0.26 | 0.27 |
| 5 | 3.5 | 10 | 0.04 | 0.33 | 0.29 |

Foreshocks for Mainshocks with $m_M \geq 6$

The ISIDE catalog presents only three $m_M \geq 6$ earthquakes, Table 3, which are here analysed separately. The L'Aquila earthquake was a Mw6 magnitude event that occurred on 6 April 2009 at 01:32 UTC in the Abruzzo region of central Italy. After BPZB declustering, we identified 540 foreshocks without considering any time or magnitude constraints. The Amatrice and Norcia events, on the other hand, were a Mw6 and Mw6.5 magnitude earthquake that occurred on 24 August 2016 and 30 October 2016, respectively. Unlike the L'Aquila earthquake, these events did not present any foreshocks. To test the effectiveness of the models, we simulated 1000 events for each of the three earthquakes using the ETAS and ETAFS models, using the magnitude of the actual mainshocks as the parent magnitude. We then calculated the distribution of the number of foreshocks for each case. We found that the ETAFS model produced significantly more foreshocks than the ETAS model. However, neither model was able to accurately describe the experimental data, as the L'Aquila earthquake had a much higher number of foreshocks than even the ETAFS model predicted. For the Amatrice and Norcia events, the ETAS model seemed to be the best fit as it did not predict any foreshocks. However, due to the limited statistics available for Mw6 magnitude earthquakes, the results are not statistically significant. Indeed, in our study, we do not consider Mw7 earthquakes because in the ISIDE catalog and in the time window considered there are no earthquakes with a magnitude greater than 7. The method has been verified for such earthquakes for the Southern California catalog [21].

**Table 3.** The Mw6 mainshock registered in the ISIDE seismic catalogue.

| Name | Mw | Date |
|---|---|---|
| L'Aquila | 6.0 | 6 April 2009 |
| Amatrice | 6.0 | 24 August 2016 |
| Norcia | 6.5 | 30 October 2016 |

## 6. Conclusions

In this study, we analyzed the Italian seismic catalog (ISIDE) and compared different types of ETAS models. The first is the standard ETAS model with the only constraint that $\alpha = b$, as proposed in [19]. The second and third variants, ETASI [30] and ETAFS [21], explicitly account for the incompleteness of the experimental catalog and a higher probability of foreshock occurrence, respectively. The ETAFS model combines the concept of bottom-up triggering for the aftershock side and top-down loading for the occurrence of foreshocks. By simulating the Italian seismicity with the ETAS model, we show that the fixed-$\alpha$ ETAS model accurately reproduces the number of aftershocks in the ISIDE catalog when the magnitude difference between mainshock and aftershocks is relatively small. However, an excess of aftershocks is observed in the ETAS model compared to the ISIDE catalog when this difference increases. We attribute this effect to the short-term aftershock incompleteness, which becomes much more important as the magnitude difference between mainshock and aftershocks increases. To address this issue, we demonstrate that introducing the incompleteness ingredient in the ETAS model (ETASI model) makes it possible to better describe the occurrence of aftershocks by removing events that were not recorded by seismic stations.

The key observation, however, is that both the ETAS model and the ETASI model fail to capture the number of foreshocks present in the ISIDE catalog. Our study clearly shows an excess of foreshocks in the ISIDE catalog for mainshocks with magnitude $m < 6$ compared to what is predicted by the ETAS and ETASI models. In contrast, the ETAFS model, which introduces a preparatory phase accompanied by foreshocks in a point process description, accurately describes the seismicity reported by the ISIDE catalog for both foreshocks and aftershocks. These patterns are consistent with those found for the Southern California seismic catalog in ref. [21], suggesting that they are a stable feature of seismic occurrences. The situation for the three mainshocks with $m > 6$ in the ISIDE catalog is different. Indeed, two of them, the Amatrice and the Norcia earthquakes, do not present foreshocks, whereas the L'Aquila earthquake presents a number of foreshocks significantly larger than those predicted by the ETAFS model.

We finally remark that the ETAFS model assumes that the magnitude of the mainshock is encoded in the spatial organization of the foreshocks. The spatial kernel of $Q_f$ is of the order of the area fractured by the mainshock. Therefore, while the prediction of a mainshock with a bottom-up triggering ETAS model is purely random, in the case of a top-down loading model, it would be possible to use the organization of foreshock epicenters as forecasting information. It is important to note that we did not assume a direct dependency between the magnitude of foreshocks and the magnitude of the mainshock, since the introduction of such dependence is still being studied [51] and is beyond the scope of this article.

**Author Contributions:** G.P. and E.L. contributed to the research, numerical results and writing of the manuscript. All authors have read and agreed to the published version of the manuscript.

**Funding:** This research received no external funding.

**Data Availability Statement:** The complete seismic catalog is available at http://terremoti.ingv.it/en/iside, accessed on 30 April 2021.

**Acknowledgments:** G.P. acknowledges support by MEXT Project for Seismology TowArd Research innovation with Data of Earthquake (STAR-E Project), Grant Number: JPJ010217. E.L. acknowledges support from project PRIN201798CZLJ and from VALERE project of the University of Campania "L. Vanvitelli". G.P. and E.L. also thank Jiancang Zhuang for helpful discussions.

**Conflicts of Interest:** The authors declare no conflict of interest.

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
