# Peer review of "Incorporating Foreshocks in an Epidemic-like Description of Seismic Occurrence in Italy"

_applsci, doi:10.3390/app13084891_

Round 1

Reviewer 1 Report

1. The text needs English editing.

2. The methodology needs more explanation. It is suggested that the steps be clarified with a flowchart.

3. Considering the "b" (the exponent of the Gutenberg-Richter law) depends on regional seismicity, where is the value origin? Is it considered according to the seismicity conditions of central Italy?

4. This method was applied to the Italian catalogs with a magnitude of less than 7. Is this method checked for stronger ground motions with a magnitude above 7 (considering the long-term reflections and refractions of the waves)? If this item requires additional studies out of the scope of this article, it is recommended to specify the range of the studied magnitude in the introduction and conclusion.

Author Response

1. The text needs English editing.
1A. We thank the referee for the comment. We have revised the text improving the English.

2. The methodology needs more explanation. It is suggested that the steps be clarified with a flowchart.
2A. We thank the referee for the comment. We have added Fig.3, which represents a flowchart of the method.

3. Considering the "b" (the exponent of the Gutenberg-Richter law) depends on regional seismicity, where is the value origin? Is it considered according to the seismicity conditions of central Italy?
3A. We thank the referee for the comment. We calculated the b-value considering the entire Italian Iside catalog from 2005/04/16 to 2021/04/30 and its value is b=0.95. This value that is the one used to perform the declustering of seismic catalogs, even if results only weaky depend on the specifc choice for the b value.

4. This method was applied to the Italian catalogs with a magnitude of less than 7. Is this method checked for stronger ground motions with a magnitude above 7 (considering the long-term reflections and refractions of the waves)? If this item requires additional studies out of the scope of this article, it is recommended to specify the range of the studied magnitude in the introduction and conclusion.
4A. We thank the referee the for the comment. There is no specific contraint for the magnitude range to consider, We do not consider magnitude greater 7 earthquakes in the present study, simply because in the Italian seismic catalog ISIDE from 2005/04/16 to 2021/04/30 there are no earthquakes with magnitude greater than 7. The method has been verified for earthquakes of magnitude greater than 7 for the Southern California catalog (Petrillo & Lippiello, 2021).
More generally, the method is expected to be more efficient the larger is the number of mainshocks to include in the analysis. For this reason, results at larger magnitude are less reliable and we have explicitly isolated the anlysis of the two M6+ earthquakes from the smaller magntudes in the present study.
In the revised version of the article we better specify this point.

Reviewer 2 Report

This study improves the ETAS model for cluster analysis and forecasting techniques to accurately predict aftershock and foreshock occurrences. Generally, the paper is well-organized. However, the following minor comments should be considered.

1.      Please explain why the functional form proposed by Kagan [39] and Helmstetter, et al. [40] was used in this study.

2.      The gap of the study (introduction section) should be described clearly.

3.      Conclusions can be shortened.

4.      The reference format in the manuscript should be checked. For example:

“Ref. [13] has also shown that foreshock activity observed in the Italian catalog is consistent with…”. It should be “Marzocchi and Zhuang [13] has also shown that foreshock activity observed in the Italian catalog is consistent with…”

5.      It should not be abbreviated without prior definitions. For example: “Sec. 1”, “Sec. 2” at the end of the introduction. Please check the whole manuscript.

6.      Figure 3 and Figure 4 are needed to improve. It is not easy to see.

7.      The paper should be rechecked for grammatical mistakes and sentence structures.

Author Response

1. Please explain why the functional form proposed by Kagan [39] and Helmstetter, et al. [40] was used in this study.
1A. The use of the functional form for $m^{th}$ (Eq.(3)) is the one which is more compatible with experimental data. Furthermore, the origin of a logarithmic deacy of $m^{th}$ (Eq.(3)) can be understood in terms of the behavior of  the seismic waveform envelope $\mu(t)$ at times $t$ following a mainshock. Specifically, $\mu(t)$ is always greater than a minimum value $\mu_c(t)$, which exhibits a logarithmic decay similar to that of $M_T(t)$. Lippiello et. al (2016) have explained the existence of $\mu_c(t)$ in terms of overlap between aftershock coda waves, and have demonstrated that the decay of $\mu_c(t)$ incorporates the parameters governing the decay of aftershocks according to the Omori-Utsu law. Consequently, it is possible to estimate the expected number of aftershocks in the immediate aftermath of a mainshock. 
In the revised version of the manuscript we clarify this point.

2. The gap of the study (introduction section) should be described clearly.
2A. In the introduction of the revised version we better clarify that previous results indicate a deficit of foreshocks in the ETAS compared to the instrumental cataogs and that the pruprpouse of this study is to show that this deficit can be compensated via the ETAFS model.

3. Conclusions can be shortened.
3A. The concluding paragraph is already of only 30 lines. We have difficulties to further shortenn it. 

4. The reference format in the manuscript should be checked. For example: “Ref. [13] has also shown that foreshock activity observed in the Italian catalog is consistent with…”. It should be “Marzocchi and Zhuang [13] has also shown that foreshock activity observed in the Italian catalog is consistent with…”
4A. We thank the referee for pointing out the inconsistency. We fixed the reference format. 

5. It should not be abbreviated without prior definitions. For example: “Sec. 1”, “Sec. 2” at the end of the introduction. Please check the whole manuscript.
5A. We thank the referee for the comment. We have removed the abbreviation Sec.1, etc. leaving only Tab. and Fig. as abbreviations.

6. Figure 3 and Figure 4 are needed to improve. It is not easy to see.
6A. We have enlarged the size of the figures in the revised version of the manuscript.

7. The paper should be rechecked for grammatical mistakes and sentence structures.
7A. We thank the referee for the comment. We have revised the text improving the English.

Reviewer 3 Report

Modeling alone can work in the first iteration in areas with pronounced, and more or less continuous, seismicity (such as southern California and Italy), but for any concrete conclusion the models should be tested in areas of moderate seismicity and/or in areas which are characterized by very irregular occurrence of earthquakes (sporadic occurrence of strong earthquakes, swarms of weaker/moderate earthquakes without a pronounced mainshock, ...).

The last sentence in the abstract describes the problem well - the seismicity of an area is characterized by numerous parameters and is highly variable. What is stated is one of the characteristics of the seismicity of an area and cannot be considered in general, as well as some other characteristics that are stated in the text as being valid in general.

Some statements are from a seismological point of view incorrect, insufficiently precise and/or quite questionable, and some are even potentially dangerous (e.g., „Essentially, any earthquake can produce a certain number of aftershocks, which are typically of smaller magnitudes, but there is also a non-negligible chance that it could trigger a larger magnitude earthquake. In the latter case, there is an increase in the seismic rate before the occurrence of the mainshock, and earthquakes responsible for this increase are usually referred to as foreshocks.“, „the aftershock incompleteness present in the experimental catalog, which is caused by the overlapping of the coda waves“, „it would in principle be possible to use the organization of foreshock epicenters as a forecasting information.“).

The text of the last chapter in the Introduction should be harmonized with the chapter numbers

The order of tables 2 and 3 should be replaced (harmonize with the indication in the text).

Author Response

1. Modeling alone can work in the first iteration in areas with pronounced, and more or less continuous, seismicity (such as southern California and Italy), but for any concrete conclusion the models should be tested in areas of moderate seismicity and/or in areas which are characterized by very irregular occurrence of earthquakes (sporadic occurrence of strong earthquakes, swarms of weaker/moderate earthquakes without a pronounced mainshock, ...).
1A. We do not understand what the reviewer intends for continuous seismic activity. In Southern California and Italy, activity is really intermittent with most of earthquakes concentarting in the few days after large earthquakes. This feature is fully captured in by the ETAS model and its variant (the ETAFS model). If the referee refers to background seismicity, this corresponds to the term $\mu(x,y)$ in Eq.(2) whic changes dramatically for region to region and for small values also leads to sporadic occurrence of strong earthquakes. But this does not affect our results that are focused on the triggring mecahnism, which is the additive term to $\mu(x,y)$ in Eq.(2). Concerning swarm activities, they are not included in our analysis and represent a different problem.

2. The last sentence in the abstract describes the problem well - the seismicity of an area is characterized by numerous parameters and is highly variable. What is stated is one of the characteristics of the seismicity of an area and cannot be considered in general, as well as some other characteristics that are stated in the text as being valid in general.
2A. The main goal of the study is to show that foreshocks occurred in the Italian regions have a prognostic value and in principle can be used for future earthquake forecasting. Obviously this result holds for the Italian region but in principle, since the mechanism for earthquake nucleation are expected to be quite universal, our study can be reforumalated to apply to other seismic regions. 

3. Some statements are from a seismological point of view incorrect, insufficiently precise and/or quite questionable, and some are even potentially dangerous (e.g., „Essentially, any earthquake can produce a certain number of aftershocks, which are typically of smaller magnitudes, but there is also a non-negligible chance that it could trigger a larger magnitude earthquake. In the latter case, there is an increase in the seismic rate before the occurrence of the mainshock, and earthquakes responsible for this increase are usually referred to as foreshocks.“, „the aftershock incompleteness present in the experimental catalog, which is caused by the overlapping of the coda waves“, „it would in principle be possible to use the organization of foreshock epicenters as a forecasting information.“).
3A. The first two quoted sentences „Essentially, any earthquake can produce a certain number of aftershocks, which are typically of smaller magnitudes, but there is also a non-negligible chance that it could trigger a larger magnitude earthquake. In the latter case, there is an increase in the seismic rate before the occurrence of the mainshock, and earthquakes responsible for this increase are usually referred to as foreshocks.“, „the aftershock incompleteness present in the experimental catalog, which is caused by the overlapping of the coda waves“ are vastly accepted features of seismic catalogs. We really do not undersand why the reviewer can consider them incorrect or even not precise. Concerning the last quoted sentence  „it would in principle be possible to use the organization of foreshock epicenters as a forecasting information.“ this is what is stated in the quoted ref.[7,8] and this is the key difference bewteen the bottom-up and top-down scenarios. Again we do not identify any problem with this statement.  

4. The text of the last chapter in the Introduction should be harmonized with the chapter numbers.
4A. We thank the referee for the comment but it seems to us that the listing of sections in the last chapter of the introduction harmonizes with the numbers in the article.

5. The order of tables 2 and 3 should be replaced (harmonize with the indication in the text).
5A. We thank the referee for pointing out this inconsistency, we have fixed the nomenclature accordingly in the revised version of the manuscript.